# Psychomotor Limitations of Overweight and Obese Five-Year-Old Children: Influence of Body Mass Indices on Motor, Perceptual, and Social-Emotional Skills

**DOI:** 10.3390/ijerph16030427

**Published:** 2019-02-01

**Authors:** Pedro Gil Madrona, Sonia J. Romero Martínez, Nieves María Sáez-Gallego, Xavier G. Ordóñez Camacho

**Affiliations:** 1Faculty of Education of Albacete, Department of Didactics of Musical Expression, Plastic Arts and Self-Expression through Movement, University of Castilla-La Mancha, 02071 Albacete, Spain; pedro.gil@uclm.es; 2Faculty of Health Sciences and Education, Department of Psychology, Madrid Open University, 28400 Madrid, Spain; 3Faculty of Education of Toledo, Department of Didactics of Musical Expression, Plastic Arts and Self-Expression through Movement, University of Castilla-La Mancha, 45071 Toledo, Spain; nieves.saez@uclm.es; 4Faculty of Education, Department of Research and Psychology in Education, Complutense University of Madrid, 28040 Madrid, Spain; xavor@ucm.es

**Keywords:** obesity, body mass index, psychomotor development, motor skills, perceptual skills, social-emotional skills

## Abstract

The present research aimed to study the psycho-motor performance of five-year-old children with different body mass indices (BMI). A total of 694 pre-school children in the province of Albacete-Spain participated. Their performance in motor, perceptual, and social-emotional skills was analyzed using a standardized observation sheet (Checklist of Psychomotor Activities—CPA) and then compared according to their BMI using non-parametric statistical methods (Mann-Whitney test). Separate comparisons were made for girls and boys. Results indicated significant differences in performance amongst the groups of girls in all the motor and perceptual activities, and in the social relationships component of the social-emotional factor. These differences seemed to penalize motor activities, perceptual skills, and social relationships in overweight and obese girls compared to normal weight girls. In the case of boys, there were significant differences in laterality and visual-motor coordination (favoring overweight boys). Differences in respiratory control were also found, but in this case, penalizing obese boys compared to normal weight boys. Knowledge of possible psychomotor limitations in obese children could allow psychologists and healthcare professionals to design more focused interventions.

## 1. Introduction

Child development is a multidimensional evolutionary process through which human beings acquire increasing autonomy until they reach adulthood. During the first years of life, this development is rapid and very important because it lays the foundation for future learning; it is also closely linked to the quality and quantity of stimuli that children receive during this period. Preschool education plays a crucial role in this development, since it is the first stage of institutionalized learning [1,2].

Psychomotricity is an important aspect of child development; therefore, it should be taken into account at all levels of basic education, especially in preschool. Psychomotricity studies the relationship between movement and cognitive function, investigates the importance of movement in personality development and learning, and integrates physical, emotional, and cognitive variables [3].

Therefore, preschool education should focus on the acquisition of physical, emotional, and cognitive skills through activities which allow children to exercise, gain self-confidence, autonomy, and initiative. The main purpose of this educational period is to foster the physical, emotional, and intellectual development of the child, giving special relevance to knowledge, appreciation, and control of their own bodies [4]. Another purpose of this stage of learning is to nurture the child’s understanding of their potential and to encourage them to use available resources with autonomy and independence. Furthermore, “health education” is also important during this period, which should include activities aimed to acquire, choose, and maintain healthy habits and behaviors [5,6].

Physical education, especially when practice is structured [7], has been proved to be an effective means of achieving these objectives [8,9,10] since it stimulates a holistic development of the child in terms of physical, cognitive, and social-emotional aspects [11]. Physical education can optimize the development of body schema, perceptual-motor skills, postural hygiene, emotional expression, creativity, social skills [1], attention [12], and cognitive processes [13], as well as the construction of the child’s personality and behavior [14]. These results have been recognized by the different actors in the education system, such as parents, kindergarten teachers, and educators, education administrators, health professionals, and students, who have had the opportunity to practice physical education [15]. Strategies for psychomotor teaching begin with planning classes and looking for students to develop fine and gross motor skills, as well as cognitive and emotional abilities [16,17]. Methods are incorporated that teach coordination, balance, and the manipulation of objects and materials to represent and create images (symbols), and learning through game-playing [18].

The components of psychomotor development are: (a) tone: forms part of the body schema and includes muscle tension and tonic postural control. It is a permanent source of stimuli, and training in this area may help pre-school children learn to control their emotions and behavior; (b) body schema: corresponds to the organization of sensations in relation to the body. It can be developed in the teaching-learning process through body awareness gained by performing different motor activities, such as imitation and exploration. The body schema is a key feature in the organization of the personality because it contributes to the relationship of different aspects of the same to the context; (c) laterality: related to the lateral dominance of eyes, hands, and feet. In preschool education, it can be developed through body movements that reinforce the concepts of right and left; (d) temporal-spatial orientation: related to the temporal and spatial organization in daily life. It is fundamental to the psychomotor development of children because it allows them to understand relationships between objects, persons and actions; and (e) coordination: the set of motor activities and abilities that involve interactions between the muscular, skeletal, and nervous systems in daily life. In preschool education, children may be trained by performing exercises such as alternating movements of feet and arms, singing, following instructions, making facial gestures, etc. Coordination is important for psychomotor development since it involves the integration of perceived data in order to perform complex physical tasks [19,20].

There are two main theories that explain psychomotor development [21]; the psychokinetic method of Jean Le Boulch and the “psychogenetic paradigm” of Jean Piaget. The first theory considers the human being as a “psychosomatic unit” with two components: (1) a “psychic” component, referring to psychic activity, which includes cognitive and emotional factors, and (2) a “motor” component, which constitutes the motor function, or in other words, movement. This theory is centered on the relationship between psychic activity and motor activity and studies their interdependence. From this perspective, psychomotricity is founded on psychology, neurophysiology, psychiatry, and psychoanalysis and is, therefore, a comprehensive approach to education or therapy aimed at developing the capabilities of the person through movement. The second theory proposed by Jean Piaget, the “psychogenetic paradigm”, is based on a particular concept of intelligence that considers mental structures as organizational properties of intelligence that are formed during ontogeny by the effect of natural and spontaneous maturation. This paradigm is founded on two basic mechanisms of development: assimilation and accommodation.

Regarding overweight and obesity, they are related to environmental and socio-economic factors, lifestyle preferences and cultural environment [22,23,24]. After comparing the psycho-motor development of 58 children aged 4 to 4.5 years according to nutritional status, Handal et al. [25] emphasized the importance of raising awareness in the educational community of the problems generated by childhood obesity, focusing on reducing sedentary behavior. In this study, the preschoolers with obesity and overweight showed a poorer motor profile compared to their normal weight peers.

Indeed, obesity is an issue that must be taken seriously. Being Overweight in adults has been shown to lead to early-onset health problems [26], decrease quality of life and increase risk of diseases such as type 2 diabetes, certain types of cancer, sleep apnea, depression, and cardiovascular disease [27]. In addition, overweight and obesity can lead to psycho-motor limitations, which result in an increase in social discrimination, low self-esteem, and depression [28]. These emotional maladjustments in youth, during physical activities practice, encourage the group to choose more individualistic and sedentary activities, which reduce opportunities for optimal psychomotor development and the reduction of obesity [28,29,30].

Overweight and obesity also have effects on social-emotional skills. Peralta et al. [31] studied the consequences of obesity on perception of ability and body image in adolescents, finding that obese participants felt less athletic, agile, fast, and fit; overall, they had poorer self-image perception than normal weight participants. In the same way, Utesh et al. [32] have recently found that physical self-concept is a moderate predictive factor of physical activity in a sample of adolescents.

Most of the studies reviewed centered on consequences of being overweight in motor competence, cognition, perception and/or emotional development in adolescents; therefore, the main objective of the present study is to compare the psychomotor development and social-emotional aspects of five-year-old children according to their BMI. This age was chosen because it is a pivotal stage between early childhood education and primary education, when other psychomotor skills are acquired. Furthermore, some studies have reported differences in psychomotor development according to gender [33,34,35]. Preschool girls were found to have more highly developed verbal language and fine motor skills, but there was no consensus regarding gross motor skills. This leads to the second objective of this study, which is to analyze whether there are differences between boys and girls regarding the influence of BMI on gross motor skills, perceptual skills, and social-emotional abilities.

Our starting hypothesis is that overweight and malnourished children (according to their BMI) will perform worse on psychomotor tasks than normal weight children and that these differences will affect girls and boys differently. The results of the present work could allow identification of factors influenced by BMI, enabling the design of both preventive and corrective interventions for use by health professionals to foster the holistic and harmonious development of the child.

## 2. Materials and Methods

### 2.1. Participants

Information was obtained from 694 five-year-old children enrolled in the third year of early childhood education. Data collection was carried out by the teachers of 32 groups in 11 schools in the province of Albacete, Spain, selected at random from a pool of 50 schools (35 public and 15 private) located in Albacete during the year 2016 according to the town hall web page [36]. In order to carry out the collection of information, parental consent was requested and the anonymity of the data was guaranteed. The sample was non-probabilistic, and once the schools were selected, the snowball technique was used to select the participating groups. The sample was composed of 46.7% girls and 53.3% boys. 69% of the students belonged to public schools and 65% of the sample practiced some kind of extracurricular sports activity.

Following the recommendations of the Food and Nutrition Technical Assistance (FANTA III) Project [37] on anthropometric indicators (weight and height) for evaluating overweight and obesity in pediatric contexts, the children were divided into five groups according to their BMI (malnutrition, moderate malnutrition, normal weight, overweight, and obesity).

The procedure to classify the children was as follows: first, the BMI of each child was calculated (BMI = kg/m^2^). The FANTA III project [37] provides tables on the nutritional status of children from five to 18 years of age based on percentiles obtained in the project. Then, these tables were used to classify the children in terms of nutritional status according to their BMI. Table 1 presents the classification ranges and the number of boys and girls in each group.

### 2.2. Instruments

The evaluation instrument was the Checklist of Psychomotor Activities (CPA) [38], which collects information on three factors of children’s psycho-motor development: motor, perceptual and social-emotional. The CPA is composed of 12 variables divided as follows: the physical-motor factor is evaluated by laterality items (LAT—seven tasks), dynamic coordination (DC—six tasks), balance (BAL—five tasks), motor execution (ME—three tasks) and tonic postural control (TPC—three tasks). The perceptual-motor factor measures respiratory control (RC—three tasks), schema and body image (BI—four tasks), motor dissociation (MD—three tasks), visual-motor coordination (VMC—six tasks), and spatial orientation (SO—two actions). Finally, the social-emotional factor was measured by the emotional control items (EC—six tasks) and social relations items (SR—five tasks). Each of the items is made up of the sum of values of the various tasks that compose it. The tasks contained in each of the variables can be found in Table 2.

The CPA was applied in the physical education class (in two classes, concretely: 50 min for the motor tasks and 50 min for the perceptual activities). The teacher and one member of the research team prepared the material previously for the class and then, during the class, each child was observed performing activities. In the case of motor and perceptual activities, the child was asked to carry out each task five times, and the frequency with which they performed each task correctly (1–5) was recorded. The social-emotional items were evaluated by the teacher, who scored the behavior and attitude of each child at the end of the academic term, using a Likert type scale (from 1–5, 5 being the highest score in all cases). The psychometric properties (validity and reliability) of the CPA may be consulted in Romero [38].

### 2.3. Procedure

First, the current study was reviewed and approved by the Research Ethics Committee of the University of Castilla-La Mancha. Second, the observed motor behaviors were recorded by the participating researcher and the classroom teacher. Instructions were given to teachers by a member of the research team 20 min before applying the instrument during recess; following testing, scores were awarded in a consensual manner by both the classroom teacher and the researcher. Teachers and researchers also measured height and weight of each participant. Weight was measured with a scale and height with a meter ruler mounted on the classroom wall for calculating the BMIs. Third, data were preprocessed for the analysis, which involved: (a) assumptions testing (normality), (b) analysis of outliers, and (c) classification of the children according to BMI.

### 2.4. Design and Data Analysis

A quantitative, cross-sectional, non-experimental, descriptive, and explanatory study was carried out [39]. To test the proposed hypothesis, the differences between groups formed according to BMI and gender were analyzed using Kruskal-Wallis nonparametric tests, complemented with Mann-Whitney U tests with Bonferroni corrections as post-hoc measures. We decided to use non-parametric statistics since the previous verification of assumptions using the Kolmogorov-Smirnov test indicated absence of normality for the contrast groups. The statistical package SPSS v.20 (IBM, Armonk, NY, USA) [40] was used to carry out the statistical analysis.

## 3. Results

### 3.1. Analysis of Motor Skills in the Sample of Girls

The results of the Kruskal-Wallis H test for motor skills according to BMI in girls are presented in Table 3. The Table also includes descriptive statistics indicating the lack of normality in the groups, which is why parametric statistics could not be used.

Post-hoc analysis was performed to establish pairwise comparisons (see Mann-Whitney’s U in Appendix A). This analysis indicates differences between the group of malnourished girls and the normal-weight, overweight, and obese girls in laterality, with the malnourished girls showing lower scores (see Figure 1). Regarding dynamic coordination, there are significant differences between the groups with normal weight/moderate malnutrition and obese girls. In both cases, the dynamic coordination scores are lower for the obese girls (see Figure 1). In terms of motor execution, there are significant differences between malnourished and normal-weight girls compared to overweight and obese girls. As can be seen in Figure 1, girls with higher BMI obtain worse results in motor execution. With regard to tonic postural control, there are differences between overweight/obese girls and all the other groups. Differences indicate poorer control in overweight and obese girls.

Finally, regarding motor aspects, there are differences in balance between the group of malnourished girls and the remaining groups, with low-weight girls showing better results.

### 3.2. Analysis of Motor Skills in the Sample of Boys

The results of the Kruskal-Wallis H test for motor skills according to BMI in boys are presented in Table 4. The Table also includes descriptive statistics indicating the lack of normality in the groups; therefore parametric statistics could not be used.

The post-hoc analysis (see Mann-Whitney’s U in Appendix A) indicates differences only in laterality (see Figure 2), between overweight/obese boys and the remaining groups. As can be seen in Figure 2, boys with higher BMIs obtain better scores in laterality (a similar result to that of the girls).

### 3.3. Analysis of Perceptual Skills in the Sample of Girls

The results of the Kruskal-Wallis H test for perceptual skills according to BMI in girls are presented in Table 5. The Table also includes descriptive statistics indicating the lack of normality in the groups; therefore, parametric statistics could not be used.

Post-hoc analysis (see Mann-Whitney’s U in Appendix A) indicates significant differences between groups of girls in all the variables that compose the perceptual factor, as explained below. In respiratory control, there are significant differences between overweight/obese girls and the remaining groups (see Figure 3); analysis of average ranges shows that in all cases, girls with higher BMI have lower respiratory control scores. Regarding body image, there are differences between malnourished and obese/overweight girls and also between girls with normal weight compared to obese/overweight girls (Figure 3). In all cases, mean ranges indicate lower scores in overweight and obese girls. The same occurs with the motor dissociation variable. With regard to visual-motor coordination, there are significant differences between normal-weight girls and obese/overweight girls, with normal weight girls presenting higher scores (see Figure 3). Finally, in terms of spatial orientation, girls with normal weight have higher scores than overweight girls (Figure 3).

### 3.4. Analysis of Perceptual Skills in the Sample of Boys

The results of the Kruskal-Wallis H test for perceptual skills according to BMI in boys are presented in Table 6. The Table also includes descriptive statistics indicating the lack of normality in the groups; therefore, parametric statistics could not be used.

Post-hoc analysis (see Mann-Whitney’s U in Appendix A) indicates significant differences in respiratory control and visual-motor coordination (see Figure 4) between malnourished and overweight/obese boys. Mean ranges indicate that boys with obesity and overweight have lower RC scores, but higher VC scores.

### 3.5. Analysis of Social-Emotional Skills

Table 7 and Table 8 show the results for the social-emotional factor in girls and boys, respectively. There are significant differences between the groups of girls in terms of social relationships, namely between girls of normal weight and the malnutrition group (U = 1629; Z = −2.316; *p* = 0.021; r = 0.155) and between girls of normal weight and overweight (U = 3482; Z = −2.395; *p* = 0.017; r = 0.153). In the group of boys there are no significant differences (see Table 8).

## 4. Discussion

The main objective of this study was to analyze the psychomotor performance of physical-motor, perceptual-motor and social-emotional skills in five-year-old girls and boys with different BMIs (malnutrition, moderate malnutrition, normal weight, overweight, and obesity).

Firstly, observing the results obtained according to gender, it can be concluded that there are significant differences between the BMI groups in all the physical-motor and perceptual skills in the case of girls, whereas amongst the groups of boys, differences were found in only some isolated aspects (laterality, respiratory control, and visual-motor coordination). Therefore, it can be affirmed that BMI has a stronger influence on the performance of physical-motor and perceptual-motor skills amongst girls. Specifically, girls with normal weight show better results compared to overweight and obese girls.

A priori, these results are in contrast to those obtained by Peyre et al., Flatters et al., and Toivainen et al. [32,33,34], who found no differences in gross motor ability between girls and boys. Meanwhile, other authors who analyzed motor differences between boys and girls in different educational stages and in some studies according to BMI, did find that boys displayed better motor performance [41,42]; they thought this may be due to higher levels of motor involvement compared to girls. It is possible that, in the case of girls, differences in BMI have a greater impact on psycho-motor ability because they practice less physical activity and have fewer motor experiences than boys [43,44].

If we examine physical and motor skills in detail, we can see that obese girls exhibit lower scores than their normal weight peers in more than half the tests, namely in dynamic coordination, motor performance and balance. At the same time, in tonic postural control, they obtain the second worst score, very close to the worst score (obtained by the girls with overweight). In all of the motor factor tests, the girls with normal weight and malnutrition obtained the best results. Along these lines, Bucco-Dos Santos et al. [43] found that performance of normal-weight girls was significantly better than their counterparts with obesity in variables such as gross motor skills and balance. Likewise, Laguna et al. [45] found that children between the ages of 6 and 12 with obesity presented worse results in balance and dynamic coordination. They argue that balance problems in overweight children result in poorer dynamic coordination due to lower awareness of weight distribution and center of gravity.

The results also show that girls with malnutrition perform better in some aspects, such as motor skills, tonic postural control, and balance. It is possible that these results are owing to the fact that to accomplish tasks, a lower body mass is easier to manage; therefore they are better at motor execution and controlling tone and balance. As Aleixo et al. [46] stated, obesity and excess body mass cause postural changes, reduce stability, and require an increase in biomechanical adaptations. In relation to these results, Nascimento et al. [47] also found, when comparing the chronological age and the motor-coordination age in boys and girls from nine to 11 years, that the participants with low weight (malnutrition state) presented a higher motor-coordination age compared to chronological age.

On the other hand, it is noteworthy that, in the laterality tests, the obese and overweight participants obtained the best results, above all the other groups. One possible explanation for this may lie in the lower motor skills of the groups with the highest BMI [48], requiring better lateralization in order to adapt to situational characteristics when interacting with the environment. However, this superiority of obese and overweight participants in laterality occurs in both girls and boys, which seems to indicate that there are no differences in laterality according to gender, a feature highlighted in other studies [49,50].

To conclude with the motor factor, girls with normal weight appear to be the most stable group, as stated by Bucco-Dos Santos et al. [43] and Méndez et al. [25]; they obtain the best scores in tests of dynamic coordination and in the sum of all variables. In addition, they are (1) significantly better than overweight and obese girls in motor performance, tonic-postural control and balance, (2) better than girls with obesity in dynamic coordination, and (3) better than girls with malnutrition in laterality. It is important to emphasize that possible motor deficiencies in obese children should be treated with caution and may be due to the influence of education and motor stimulation.

Secondly, with regard to perceptual aspects, it should be noted that overweight and obese boys have poorer respiratory control than their counterparts with lower BMI, but are better in visual-motor coordination. Regarding the other perceptual-motor aspects, no significant differences were obtained between the groups of boys.

In the case of girls, significant differences between groups are again found in all perceptual aspects. Overweight and obese girls score worse than girls with lower BMI in respiratory control, body image and motor dissociation. Moreover, overweight girls also present the worst scores in visual coordination and spatial orientation, contrasting with normal weight girls who achieve the best scores in both aspects. These results are in accordance once more with those obtained by Bucco-Dos Santos et al. [29]; their normal weight participants had better body schema, spatial organization and temporal organization than their counterparts with overweight or obesity. The results also coincide with those of Brum et al. [51], who affirmed that children with obesity had greater problems with body schema and temporal organization. As in the motor factor, normal weight girls are the most stable and superior group since they obtain the highest scores in some aspects (visual coordination, spatial orientation) and the second highest scores in the remaining perceptual-motor skills.

Thirdly, there are significant differences in social-emotional aspects only in the case of girls. Specifically, girls with normal weight have better social relationships than those with malnutrition or obesity. These differences may be related to factors such as low self-esteem amongst extremely thin or obese girls; this result is supported by the research of Molina et al. [43], who found that obese children scored lower on dimensions of self-esteem and quality of life related to health. The fact that, in our study, girls with obesity obtain poorer results in social relationships may encourage them to engage in individual activities, including sedentary activities, which reduces opportunities for optimal psycho-motor development and obesity reduction [28,30].

Despite the interesting results discussed above, the present study has some limitations. First, since the sample was not probabilistic, the results are not generalizable to the wider population. Additionally, the children were classified based on BMI as a single indicator; in order to achieve a richer and more accurate classification, other anthropometric, biochemical, clinical (e.g., nutritional requirements), and dietary intake measures would need to be considered. Future research in this field can be directed along these lines.

Finally, despite the large sample size, it was necessary to use non-parametric statistics due to non-compliance with the assumptions; parametric Analysis of variance could not be used to test interactions between BMI and gender. Despite this limitation, the present research has several practical implications for teachers, healthcare practitioners, and children. The results may be used to develop or re-establish children’s capacities, focusing on their potential and the different aptitudes we found to be influenced by gender and BMI profiles. Psychomotor interventions to improve the different skills contemplated here (motor, social-emotional, perceptual) must respond to schemes based on the analysis of children’s characteristics and knowledge of how overweight and obesity affect children’s psychomotor functioning. Taking these specificities into account will permit the design of both preventive and corrective interventions based on contents and methods adapted to the different groups of children. The results of the present study may be used in both educational and clinical settings (reeducation or psychomotor therapy) to foster the holistic and harmonious development of the child.

## 5. Conclusions

The main finding of this study is that, in the case of girls, there were significant differences in task performance according to BMI across all the psychomotor variables analyzed. These differences indicate poorer performance of obese and overweight girls in motor activities, perceptual skills, and social relationships compared to normal weight girls. On the other hand, in boys, differences between groups were found in only some isolated aspects, such as laterality, respiratory control, and visual-coordination. Contrary to what was expected, differences in laterality skills favored the group of overweight boys; however, overweight and obese boys presented poorer performance in respiratory control and visual-coordination tasks. These results have important practical implications, as interventions to improve physical-motor and perceptual skills can be focused on overweight and obese girls. Additionally, it is important to generate programs that facilitate social relationships in this group of girls. In the case of boys, it is important to focus on laterality training in children with low weight and on the improvement of perceptual abilities, such as respiratory control and visual coordination, in overweight boys.

Secondly, children with normal weight showed a more stable and superior motor performance, although they were surpassed by the low weight group in two variables: tonic postural control and balance. Again, this result indicates that it is essential to design health prevention programs that promote adequate weight control, seeking a balance that allows children to develop motor competence.

Finally, obese and overweight children (boys and girls) obtained higher scores in laterality; this result suggests that educational programs be designed to help overweight children use this trait to their advantage.

## Figures and Tables

**Figure 1 ijerph-16-00427-f001:**
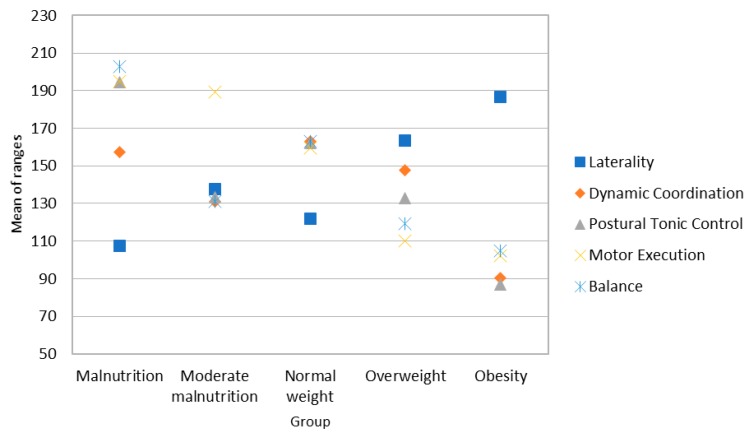
Average range of the scores for motor skills in the sample of girls.

**Figure 2 ijerph-16-00427-f002:**
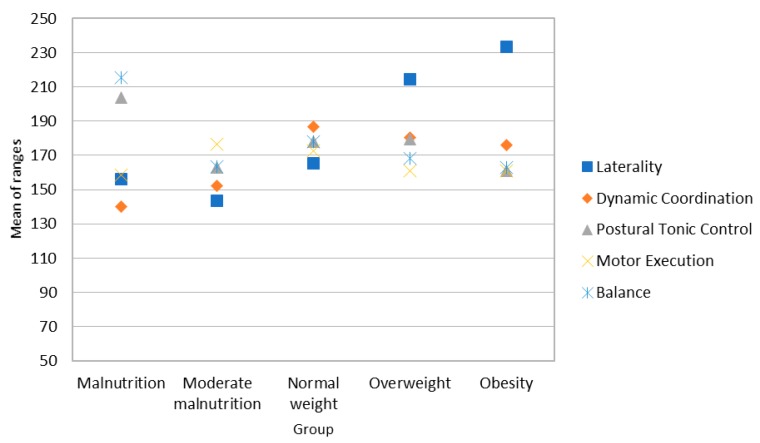
Average range of the scores for motor skills in the sample of boys.

**Figure 3 ijerph-16-00427-f003:**
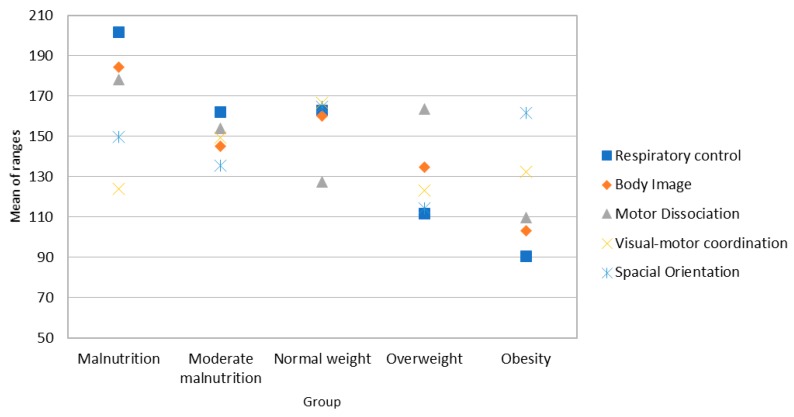
Average range of the scores for perceptual skills in the sample of girls.

**Figure 4 ijerph-16-00427-f004:**
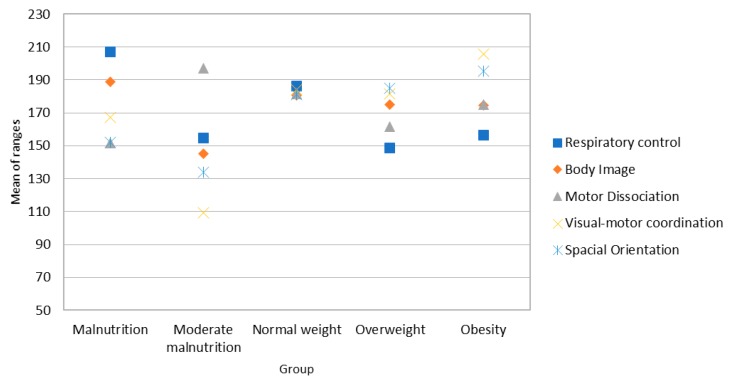
Average range of the scores for perceptual skills in the sample of boys.

**Table 1 ijerph-16-00427-t001:** Classification ranges ^1^ and number of boys and girls in each group.

	Age (Y:M)	Malnutrition < −3SD (BMI)	Moderate Malnutrition ≥ −3a < −2SD (BMI)	Normal ≥ −2a ≤ +1SD (BMI)	Overweight > +1a ≤ +2SD (BMI)	Obesity > +2SD (BMI)
Boys	5:1	Less than 12.1	12.1–12.9	13.0–16.6	16.7–18.3	18.4 or more
	5:6	Less than 12.1	12.1–12.9	13.0–16.7	16.8–18.4	18.5 or more
	*N*	36	19	210	47	43
Girls	5:1	Less than 11.8	11.8–12.6	12.7–16.9	17–18.9	19 or more
	5:6	Less than 11.7	11.7–12.6	12.7–16.9	17–19	19.1 or more
	*N*	23	22	200	45	17

^1^ Classification ranges obtained from the tables of the FANTA III Project (https://www.fantaproject.org/); BMI: body mass indices; SD: Standard deviation.

**Table 2 ijerph-16-00427-t002:** Tasks evaluated (CPA items).

Variable	Item/Task
Laterality	Grasps objects with both hands Grasps objects with the left hand Grasps objects with the right hand Hits objects with the left leg Hits objects with the right leg Throws objects with the left hand Throws objects with the right hand
Dynamic Coordination	Is able to roll on a surface Jumps with both feet together Jumps with one footIs able to move side-ways Is able to walk backwards Runs freely without difficulty
Tonic Postural Control	Moves following the indicated rhythm Runs and is able to stop at a signal Lies down flat on the back
Motor Execution	Is able to use the materials correctly Is able to jump over obstacles Is able to circle around obstacles
Balance	Maintains balance by walking in a straight line Maintains balance by walking along a curved line Maintains balance by walking on a trestle Maintains balance by walking on a bench Is able to maintain a balanced posture
Respiratory Control	Is able to hold their breath until indicated Is able to inhale for a period of time Is able to exhale according to instructions
Body Image	Recognizes body parts (hands, feet, head) Recognizes the function of body parts Recognizes own image in the mirror and in photographs Imitates movements made by the teacher or classmates
	Walks alternating arms and legs Climbs monkey bars alternating hands and legs Is able to use isolated parts of the body when instructed
Visual-Motor Coordination	Is able to manage tasks that require fine motor skills Is able to cut paper Is able to mold plasticine Catches objects in the air with both hands Throws a ball in the indicated direction Is able to bounce a ball
Spatial Orientation	Knows how to position themselves with respect to an object Performs a task in the space intended for it
Emotional Control	Expresses themselves in an appropriate way, adapting to different contexts Expresses their emotions and feelings Tries to control disproportionate expression of emotions Shows confidence when performing activities that they are capable of doing Shows interest and desire to improve Concerned about classmates who need help to perform tasks
Social Relationships	Establishes relationships according to classroom norms Shares material with classmates Respects the teacher’s instructions Shows a positive attitude when playing in a group Gets angry when loses in a game

**Table 3 ijerph-16-00427-t003:** Kruskal-Wallis test and descriptive statistics of motor skills according to BMI for the sample of girls.

	Kruskal-Wallis Test			Classification	Descriptive Statistics				
	H	df	*p*	BMI	Mean (SD)	Range	Me	G_1_	G_2_
LAT	12.646	4	0.013 *	Malnutrition	23.96 (3.52)	107.35	23	1.892	4.262
				Moderate malnutrition	25.14 (4.13)	137.36	24	0.161	−1.194
				Normal weight	27.14 (5.00)	121.80	25	0.330	−1.386
				Overweight	27.33 (5.16)	163.29	27	−0.127	−1.125
				Obesity	28.53 (3.06)	186.47	29	−0.704	0.052
DC	13.022	4	0.011 *	Malnutrition	26.87 (3.95)	157.52	27	−1.951	3.915
				Moderate malnutrition	26.32 (2.93)	131.25	26	−0.195	−0.647
				Normal weight	27.41 (2.89)	162.91	29	−1.260	−1.201
				Overweight	26.36 (4.36)	147.78	29	−1.158	1.034
				Obesity	25.24 (2.36)	90.32	25	−0.739	−0.572
TPC	22.596	4	0.000 *	Malnutrition	13.79 (2.11)	194.57	15	−2.762	8.405
				Moderate malnutrition	13.89 (1.51)	133.30	14	−0.887	−0.036
				Normal weight	13.93 (1.47)	162.07	15	−1.690	3.375
				Overweight	14.26 (1.68)	132.86	14	−1.096	0.362
				Obesity	13.55 (1.47)	86.94	13	−0.519	−0.564
ME	27.708	4	0.000 *	Malnutrition	13.87 (1.86)	195.04	15	−1.953	3.737
				Moderate malnutrition	13.82 (1.65)	189.23	15	−1.495	2.052
				Normal weight	13.16 (1.98)	159.67	14	−0.790	−0.303
				Overweight	11.87 (2.33)	110.12	12	−0.383	−0.278
				Obesity	12.00 (1.36)	102.38	12	0.166	−0.082
BAL	23.643	4	0.000 *	Malnutrition	23.04 (3.73)	203.04	25	−2.410	6.183
				Moderate malnutrition	20.36 (4.11)	131.09	20	−0.513	−0.419
				Normal weight	21.78 (3.41)	162.84	20	−0.783	−0.011
				Overweight	20.18 (3.35)	119.34	20	−0.116	−0.726
				Obesity	19.82 (2.09)	105.00	20	−0.426	0.584
SUM	12.345	4	0.015 *	Malnutrition	102.00 (9.70)	158.98	104	−1.894	4.451
				Moderate malnutrition	99.18 (8.57)	124.27	100.5	−0.939	1.078
				Normal weight	103.41 (10.28)	164.79	105	−0.512	0.317
				Overweight	99.02 (12.86)	136.22	103	−1.215	1.526
				Obesity	98.59 (6.06)	105.82	100	−0.843	2.035

* Asterisks indicate significant differences between some groups. LAT = laterality, DC = dynamic coordination, TPC = tonic postural control, ME = motor execution, BAL = balance, SUM = total score on the physical-motor factor, H = Kruskal-Wallis H test. SD = standard deviation, Me = median, G_1_ = skewness, and G_2_ = kurtosis.

**Table 4 ijerph-16-00427-t004:** Kruskal-Wallis test and descriptive statistics of motor skills according to BMI for the sample of boys.

	Kruskal-Wallis Test			Classification	Descriptive Statistics				
	H	df	*p*	BMI	Mean (SD)	Range	Me	G_1_	G_2_
LAT	26.005	4	0.000 *	Malnutrition	25.72 (5.39)	156.06	23	0.327	0.148
				Moderate malnutrition	25.42 (4.61)	143.55	24	0.492	−0.597
				Normal weight	26.40 (5.17)	165.35	24	0.149	−0.482
				Overweight	28.94 (4.88)	214.71	30	−0.421	−1.042
				Obesity	29.63 (4.11)	233.27	30	−0.447	−0.513
DC	0.021	4	0.091	Malnutrition	25.75 (3.72)	139.96	26	−0.576	−0.507
				Moderate malnutrition	25.79 (4.42)	152.45	27	−1.026	0.016
				Normal weight	27.28 (3.28)	186.70	28	−1.401	1.609
				Overweight	27.45 (2.70)	180.33	29	−1.005	−0.038
				Obesity	27.23 (2.73)	176.13	28	−0.513	−1.245
TPC	4.470	4	0.346	Malnutrition	13.89 (2.08)	203.89	15	−1.919	2.663
				Moderate malnutrition	13.21 (2.50)	162.95	14	−1.661	2.638
				Normal weight	13.76 (1.73)	178.05	14.5	−1.786	3.477
				Overweight	13.96 (1.21)	179.47	14	−0.827	−0.386
				Obesity	13.81 (1.05)	161.10	14	−0.252	−1.221
ME	5.814	4	0.213	Malnutrition	13.28 (1.68)	203.89	15	−1.877	3.311
				Moderate malnutrition	11.89 (2.58)	158.53	13	1.382	1.451
				Normal weight	12.80 (2.17)	176.71	13	−0.866	0.231
				Overweight	12.85 (1.84)	172.43	13	−0.644	−0.192
				Obesity	12.86 (1.68)	161.10	13	−0.303	−1.133
BAL	6.710	4	0.152	Malnutrition	22.17 (4.23)	215.35	25	−1.643	1.143
				Moderate malnutrition	20.89 (3.74)	163.55	21	−0.164	−1.685
				Normal weight	21.34 (3.61)	178.15	22	−0.897	0.739
				Overweight	21.45 (2.91)	168.52	22	−0.515	−1.058
				Obesity	21.14 (2.92)	162.76	21	−0.306	0.965
SUM	5.008	4	0.287	Malnutrition	100.81 (13.13)	169.68	102	−1.472	3.339
				Moderate malnutrition	97.21 (14.79)	145.18	103	−0.874	0.113
				Normal weight	101.59 (11.36)	175.24	105	−1.082	2.658
				Overweight	104.54 (8.78)	200.20	103	−0.759	1.658
				Obesity	104.67 (8.18)	188.67	103	0.466	−0.848

* Asterisks indicate significant differences between some groups. LAT = laterality, DC = dynamic coordination, TPC = tonic postural control, ME = motor execution, BAL = balance, SUM = total score on the physical-motor factor, H = Kruskal-Wallis H test. SD = standard deviation, Me = median, G_1_ = skewness, and G_2_ = kurtosis.

**Table 5 ijerph-16-00427-t005:** Kruskal-Wallis test and descriptive statistics for perceptual skills according to BMI for the sample of girls.

	Kruskal-Wallis Test			Classification	Descriptive Statistics				
	H	df	*p*	BMI	Mean (SD)	Ranges	Me	G_1_	G_2_
RC	30.757	4	0.000 *	Malnutrition	14.09 (2.13)	201.57	15	−2.534	5.951
				Moderate malnutrition	13.14 (2.69)	161.89	14.5	−1.812	2.962
				Normal weight	13.30 (2.25)	162.61	14.5	−1.440	1.830
				Overweight	11.82 (2.72)	111.56	12	−0.326	−1.300
				Obesity	11.76 (1.60)	90.53	12	0.021	0.069
BI	16.380	4	0.003 *	Malnutrition	18.27 (3.25)	184.30	20	−4.641	1.890
				Moderate malnutrition	18.74 (2.48)	145.09	20	−2.108	4.031
				Normal weight	19.22 (2.28)	160.17	20	−0.179	2.900
				Overweight	18.44 (2.75)	134.70	20	−2.119	3.633
				Obesity	18.35 (1.49)	103.03	18	−0.184	−1.498
MD	18.730	4	0.00 *	Malnutrition	13.78 (2.27)	178.20	15	−2.470	6.103
				Moderate malnutrition	13.23 (2.52)	153.83	14	−1.883	3.327
				Normal weight	13.70 (1.82)	127.47	15	−1.117	0.640
				Overweight	12.42 (2.17)	163.49	12	−0.559	−0.104
				Obesity	13.18 (1.46)	109.53	14	−1.016	1.063
VC	13.416	4	0.009 *	Malnutrition	25.26 (3.34)	123.78	25	−0.559	1.348
				Moderate malnutrition	25.73 (4.26)	149.14	26.5	−1.593	0.491
				Normal weight	26.78 (3.25)	166.74	27	0.313	5.658
				Overweight	24.93 (3.73)	123.29	26	−0.951	0.498
				Obesity	25.88 (2.38)	132.59	27	−0.268	−1.188
SO	16.404	4	0.003	Malnutrition	9.30 (0.87)	149.89	9	−1.116	0.582
				Moderate malnutrition	8.91 (1.45)	135.39	9	−1.984	4.194
				Normal weight	9.47 (1.54)	164.82	10	5.599	6.251
				Overweight	8.56 (1.86)	114.24	9	−2.301	5.851
				Obesity	9.53 (0.51)	161.59	10	−0.130	−2.267
SUM	22.006	4	0.000 *	Malnutrition	81.65 (9.06)	161.72	84	−2.947	1.043
				Moderate malnutrition	79. 27(13.00)	152.82	82	−2.179	4.677
				Normal weight	82.46 (8.45)	167.64	84	−0.309	6.448
				Overweight	76.18 (10.78)	109.21	79	−1.201	1.051
				Obesity	78.71 (3.29)	103.24	79	−0.859	1.999

* Asterisks indicate significant differences between some groups. RC = respiratory control, BI = body image, MD = motor dissociation, visual-motor coordination (VC), SO = spatial orientation, SUM = total score on the physical-motor factor, H = Kruskal-Wallis H test. Me = median, G_1_ = skewness, and G_2_ = kurtosis.

**Table 6 ijerph-16-00427-t006:** Kruskal-Wallis test and descriptive statistics for perceptual skills according to BMI for the sample of boys.

	Kruskal-Wallis Test			Classification	Descriptive Statistics				
	H	df	*p*	BMI	Mean (SD)	Range	Me	G_1_	G_2_
RC	11.995	4	0.017 *	Malnutrition	13.31 (2.86)	206.99	15	−1.551	1.430
				Moderate malnutrition	12.68 (2.16)	154.76	13	−0.207	−1.551
				Normal weight	13.28 (2.19)	186.15	14	−1.312	1.097
				Overweight	12.70 (1.95)	148.43	13	−0.874	0.753
				Obesity	12.84 (2.06)	156.51	13	−0.799	−0.062
BI	3.620	4	0.460	Malnutrition	18.50 (3.29)	188.78	20	−2.350	4.933
				Moderate malnutrition	17.84 (3.20)	145.34	20	−1.554	1.415
				Normal weight	18.99 (2.19)	180.44	20	−3.225	1.588
				Overweight	19.17 (1.30)	175.23	20	−1.487	1.353
				Obesity	19.01 (1.62)	174.51	20	−1.659	1.988
MD	4.482	4	0.345	Malnutrition	13.42 (2.77)	151.58	15	−1.845	2.655
				Moderate malnutrition	13.00 (2.44)	196.89	14	−1.791	3.179
				Normal weight	13.57 (1.94)	181.48	14	−2.194	6.188
				Overweight	13.60 (1.39)	161.48	14	−1.731	1.941
				Obesity	13.81 (1.29)	174.93	14	−1.579	2.584
VC	20.535	4	0.000 *	Malnutrition	23.72 (4.03)	167.32	24	0.021	−1.013
				Moderate malnutrition	25.05 (5.67)	109.34	28	−1.187	0.181
				Normal weight	26.45 (3.97)	184.26	28	−1.724	3.549
				Overweight	26.96 (2.58)	181.57	27	−1.287	1.847
				Obesity	27.65 (2.09)	205.56	28	−0.641	−0.606
SO	9.519	4	0.050	Malnutrition	8.83 (1.64)	152.15	9	−1.951	3.888
				Moderate malnutrition	8.37 (2.11)	134.00	9	−1.768	3.586
				Normal weight	9.27 (1.18)	181.28	10	−2.085	5.246
				Overweight	9.49 (0.65)	184.87	10	−0.928	−0.190
				Obesity	9.53 (0.76)	195.56	10	−1.950	3.956
SUM	8.613	4	0.072	Malnutrition	77.78 (11.76)	143.90	80	−1.658	2.525
				Moderate malnutrition	76.95 (11.08)	149.63	82	−1.257	0.519
				Normal weight	81.66 (9.71)	188.22	85	−2.032	4.858
				Overweight	81.91 (5.18)	163.99	83	−0.470	−0.562
				Obesity	82.86 (5.72)	184.50	83	−0.346	−0.870

* Asterisks indicate significant differences between some groups. RC = respiratory control, BI = body image, MD = motor dissociation, VC = visual-motor coordination, SO = spatial orientation, SUM = total score on the physical-motor factor, H = Kruskal-Wallis H test. SD = standard deviation, Me = median, G_1_ = skewness, and G_2_ = kurtosis.

**Table 7 ijerph-16-00427-t007:** Kruskal-Wallis test and descriptive statistics for social-emotional skills according to BMI for the sample of girls.

	Kruskal-Wallis Test			Classification	Descriptive Statistics				
	H	df	*p*	BMI	Mean (SD)	Range	Me	G_1_	G_2_
EC	7.887	4	0.096	Malnutrition	25.87 (2.91)	126.54	27	−0.983	1.044
				Moderate malnutrition	26.23 (3.72)	150.07	27.5	−2.658	3.027
				Normal weight	26.73 (3.64)	161.02	27.5	−0.427	4.921
				Overweight	25.13 (4.49)	130.38	27	−1.177	0.890
				Obesity	27.29 (2.95)	176.21	28	−2.125	5.690
SR	11.459	4	0.022	Malnutrition	19.70 (2.72)	120.67	20	−1.294	4.130
				Moderate malnutrition	20.95 (2.53)	166.59	21.5	−0.754	0.232
				Normal weight	21.04 (2.92)	164.19	21	1.162	1.577
				Overweight	19.67 (3.79)	129.57	20	−1.361	3.221
				Obesity	20.00 (2.29)	127.56	21	0.177	−1.140
SUM	9.454	4	0.051	Malnutrition	45.57 (5.18)	122.39	47	−1.252	3.405
				Moderate malnutrition	47.18 (5.69)	158.98	47.5	−1.594	2.822
				Normal weight	47.77 (6.03)	163.33	49	0.484	1.459
				Overweight	44.80 (7.53)	128.18	47	−1.048	0.715
				Obesity	47.29 (4.22)	151.59	47	−1.146	1.782
ME	27.708	4	0.000 *	Malnutrition	13.87 (1.86)	195.04	15	−1.953	3.737
				Moderate malnutrition	13.82 (1.65)	189.23	15	−1.495	2.052
				Normal weight	13.16 (1.98)	159.67	14	−0.790	−0.303
				Overweight	11.87 (2.33)	110.12	12	−0.383	−0.278
				Obesity	12.00 (1.36)	102.38	12	0.166	−0.082

* Asterisks indicate significant differences between some groups. RC = respiratory control, BI = body image, MD = motor dissociation, visual-motor coordination (VC), SO = spatial orientation, SUM = total score on the physical-motor factor, H = Kruskal-Wallis H test. SD = standard deviation, Me = median, G_1_ = skewness, and G_2_ = kurtosis.

**Table 8 ijerph-16-00427-t008:** Kruskal-Wallis test and descriptive statistics for social-emotional skills according to BMI for the sample of boys.

	Kruskal-Wallis Test			Classification	Descriptive Statistics				
	H	df	*p*	BMI	Mean (SD)	Range	Me	G_1_	G_2_
EC	6.014	4	0.096	Malnutrition	25.16 (4.69)	146.63	25	−1.354	1.808
				Moderate malnutrition	26.69 (2.31)	156.64	28	−2.040	5.638
				Normal weight	26.07 (4.26)	181.58	28	−1.886	4.437
				Overweight	26.32 (3.29)	177.16	27	−1.366	1.922
				Obesity	27.16 (2.29)	197.09	28	−0.357	−1.301
SR	5.336	4	0.022	Malnutrition	19.69 (2.82)	143.33	20	−1.077	1.747
				Moderate malnutrition	20.32 (3.66)	165.37	20	−0.939	2.181
				Normal weight	20.60 (3.14)	184.40	21	−1.269	3.118
				Overweight	20.91 (2.23)	180.01	21	0.368	−0.720
				Obesity	20.84 (2.01)	179.17	21	0.179	−2.282
SUM	5.594	4	0.051	Malnutrition	45.03 (6.05)	142.97	45	−1.314	2.831
				Moderate malnutrition	45.47 (7.99)	163.29	45	−1.031	−1.576
				Normal weight	46.67 (6.81)	183.15	49	−1.845	4.751
				Overweight	47.23 (4.44)	178.15	47	−0.587	0.241
				Obesity	48.00 (3.59)	188.49	48	−0.006	−0.936
ME	27.708	4	0.000 *	Malnutrition	13.87 (1.86)	195.04	15	−1.953	3.737
				Moderate malnutrition	13.82 (1.65)	189.23	15	−1.495	2.052
				Normal weight	13.16 (1.98)	159.67	14	−0.790	−0.303
				Overweight	11.87 (2.33)	110.12	12	−0.383	−0.278
				Obesity	12.00 (1.36)	102.38	12	0.166	−0.082

* Asterisks indicate significant differences between some groups. EC = emotional control, SR = social relationships, SUM = total score on the physical-motor factor, H = Kruskal-Wallis H test. SD = standard deviation, Me = median, G_1_ = skewness, and G_2_ = kurtosis.

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
