# Peer review of "Psychomotor Limitations of Overweight and Obese Five-Year-Old Children: Influence of Body Mass Indices on Motor, Perceptual, and Social-Emotional Skills"

_ijerph, 2019, doi:10.3390/ijerph16030427_

Round 1
Reviewer 1 Report
The authors report the results of a study on the association between weight status and psychomotor limitations in children aged 5 years living in Spain.
The work seems very interesting and the results could be used to improve specific interventions in obese children with psychomotor problems.
However the manuscript must be significantly improved in many points.In particular, in the introduction many sentences should be supported by a greater number of references with international validity. I have indicated some suggestions but more references could be added. The results should be better described, specifying in which groups significant evidence were highlighted. The conclusions must clearly and concisely report the main findings. English should be also improved.
All revisions are indicated as notes in the attached pdf.

Dear reviewer, thank you very much for the careful review of the article and the valuable suggestions made. Below you will find our answer to each point suggested.
Point 1:In the introduction many sentences should be supported by a greater number of references with international validity. I have indicated some suggestions but more references could be added.
Response 1: all the indicated references were added, and additionally we have added some more references at an international level.
Point 2:The results should be better described, specifying in which groups significant evidence were highlighted. The conclusions must clearly and concisely report the main findings.
Response 2: all the results were better described and the significant evidence of differences were highlighted. In addition, the conclutions (specially the first paragraph) were changed in order to clearly and precisely report the main findings.
Point 3: English should be also improved.
Response 3: English was improved throughout the manuscript
Point 4:All revisions are indicated as notes in the attached pdf.
Response 4: All the revisions indicated in the PDF were made
Reviewer 2 Report
Revision of the manuscript IJERPH-403332-peer-review-v1
Comments to the authors:
This is an interesting study which purpose is to compare the psychomotor development and affective-relational aspects of 5-year-old children according to their BMI. In this paper the authors have the opportunity to clarify one main conceptual discussion in relation to the so called psyco-motor development. What are the foundations, definition and implementation of psyco-motor development. As far as this reviewer knows this term has mainly accepted because the Spanish curriculum in preschool refers to it. It is well known and accepted into the Latin-american and Spanish literature, however, it is not clear enough how this concept drives with others that are studied extensively (e.g., motor, cognitive and/or affective development).
Actually, the introduction section is mainly based on evidences published in non-impact journal (mainly in Spanish), so limited quality of the evidences could be considered. It would be recommended the authors provide more evidences supporting the content under study.
It is recommended the authors revise systematic review evidence about the topic under study. In addition, there are previous studies Spanish preeschoolers into the context of psyco-motor development. See for instance:
Han et al. (2018). Effectiveness of exercise intervention on improving fundamental movement skills and motor coordination in overweight/obese children and adolescents: A systematic review published in Journal of Science and Medicine in Sport.
Terrón et al. (2018). Ecological correlates of Spanish preschoolers’ physical activity during school recess published in European Physical Education Review
Specific comments.
Abstract
Line 18. Please, state the sense of the differences so that the reader does not have to check the table to know which one of the groups presents the higher value.
Introduction
Line 39. Please, provide evidence support to this statement.
Line 52-65. The authors indicate different factors (i.e., SES, nutritional conditions and obesity) influencing children's psycho-motor development. The question is, why did the authors only focused on the obesity. Why did not the authors assessed the influence of SES and obesity together? Alternatively, why did not the authors adjust the analyses according to the SES?
Line 66 (purpose). In relation to the main purpose of this manuscript, it is well addressed the importance of educational community in regards to binomioum sedentary behaviour-nutrition with consequences in children's weight status. However, it would be needed to enhance its rationale in consequences such as motor competence, cognition, perception and/or affective development.
In addition, why do the authors just analysed 5-years-old children? Why do not they analyse 3- 4- and 5-years-old?
In addition, in the current study, comparative analyses according to not only the BMI but also children's sex (5x2 model) are conducted. However, the rationale of the effect of sex in children's psycho-motor development is not clear enough. The question is, why do the authors assume there is differences between 5-years-old boys and girls.
Methods
Line 85. For those who are not familiar with the methods used, which anthropometric indicators are required and who do you implement it? Please, clarify.
Line 86. According to Table 1, this method seems to be based on descriptive values (SD). So, please, justify why having this method is more appropriate than BMI percentiles (adjusted for sex and age) with classification in normal, overweight and obese children (a method suggested by the CDC, 2000).
Table 1. The are acronyms without explanation in tables. For instance, in table 1, it seems the authors refer to IMC for BMI. If so, please, change "IMC" by "BMI". Check it throughout the manuscript.
Line 92-105. It can be accepted the CPA has been validated previously. However, it could be also argued why the authors consider a functional aspect such as respiratory control as perceptive. In addition, taking into account that the affective-relational dimension involves just emotional control and social relationships, why do not the authors separate these two aspects in two different dimensions: a) affective (by emotional control) and b) relationships (by social relationships)? These are examples of the necessity of describing this instrument thoroughly.
In addition, please, note that the explanation of the tasks assessed is rather ambiguous. It seems this is a proxy-report by the preschool teachers in relation to the motor, perceptive and affective children's development. As this is a new instrument, please, clarify. In the case, this is a proxy report, how do the authors assess children actual motor, perceptive and affective/relational development? The original reference neither explain it (Romero-Martínez et al. (2018). Perceptual and Motor Skills, 125(6) 1070–1092).
Line 105. It seems the authors are using previous data published in the manuscript about the validity and reliability of the CPA scale with exact the same comparisons and method in Discriminant Validity (Romero-Martínez et al., 2018). If so, what is the novelty of the current study. Also, it should be stated. In the case, I am not right, please, clarify.
Line 109. How were the classroom teachers instructed? A thorough description of this procedure is requested.
Line 112. How did the researchers and teachers measure children's height and weight?
Line 116. What do the authors mean when they say ex-post facto prospective simple design? As far as this reviewer recognise (according to the procedure explained) this is a cross-sectional study. Is there any further information that should be considered/included to define the study design?
Discussion
Line 123. In terms of what are the authors comparing the results of the youngest children? Please, clarify.
Line 127. The explanation of the association of BMI and balance in young children (4-6 years-old) does not seem to be congruent enough. The authors state that in early childhood the BMI is not a factor that influence balance, however, they use data by Dinkel with babies of 5-6 months. I would suggest the authors delete the explanation in lines 128-132 by using lines 133-137 for explaining the absence of differences between normal and overweight early children.
Line 131. Why do the authors use LISREL and Prelis in the data analysis? Did the authors include any structural equation modelling for the analyses? Please, clarify.
Line 132. I mainly agree the authors justification of non-parametric analyses. However, taking into account the 5x2 comparison in terms of BMI and sex. Why did not the authors conduct a mix-model with these two factors included so that main and interaction effects could be test? This would enhance the potential of the analyses.
Results
Table 2. Please, remove the meaning from the text and clarify what does Me, G1 and G2 mean in the note below the table.
Moreover, why do the authors include an asterisk into brackets? What does this asterisk mean?
Line 135. I strongly recommend the authors to use a Figure to provide pairwise comparisons. The statistics of the post-hoc analysis could be reported in a supplemental file. Why do the authors provide the r-value? Is this for indicating the effect size? If so, how do the authors interpret this statistic? In addition, why do not the authors use another statistics such as V-value for non-parametric statistics?
Discussion
Taking into account there are so many comparative output from data (at least values from 16 dependent variables were compared according to 5 levels of BMI for boys and girls) to be discussed, and the structure and content of the discussion, why do not authors just provide values and comparisons from the SUM of every aspect in the three dimension (motor, perceptive and affective) of the CPA scale?
Line 265. Please, add "for boys and girls" at the end of the sentence.
Lines 271-277. The authors are discussing the results for boys and girls, and for obese and non-obese interchangeable. As a result, sometimes it is not clear whether they refer to the factor sex or the BMI. It would be recommended that they clearly separate the discussion of both factors.
Line 279. Lower than who? Please, clarify.
Line 282. Please, note that this is the first time the acronym (PhM) appears in the text.
Conclusions
Line 352. Please, rewrite the first conclusion. The sentence is rather large and requires improvement.
Line 359. This is not a conclusion derived from the results of the study directly. This is a secondary assumption by the authors that has been already stated in the discussion section. It is also repeated in the third conclusion.
Author Response
Dear reviewer, thank you very much for the careful review of the article and the valuable suggestions made. Below you will find our answer to each point suggested.
Point 1:In this paper the authors have the opportunity to clarify one main conceptual discussion in relation to the so called psyco-motor development. What are the foundations, definition and implementation of psyco-motor development. As far as this reviewer knows this term has mainly accepted because the Spanish curriculum in preschool refers to it. It is well known and accepted into the Latin-american and Spanish literature, however, it is not clear enough how this concept drives with others that are studied extensively (e.g., motor, cognitive and/or affective development).
Response 1: Paragraphs 2 to 7 of the introduction were rewritted to include a conceptual discussion about definition, implementation, theories and components of psychomotricity and their relationships.
Point 2: Actually, the introduction section is mainly based on evidences published in non-impact journal (mainly in Spanish), so limited quality of the evidences could be considered. It would be recommended the authors provide more evidences supporting the content under study.
It is recommended the authors revise systematic review evidence about the topic under study. In addition, there are previous studies Spanish preeschoolers into the context of psyco-motor development. See for instance:
Han et al. (2018). Effectiveness of exercise intervention on improving fundamental movement skills and motor coordination in overweight/obese children and adolescents: A systematic review published in Journal of Science and Medicine in Sport.
Terrón et al. (2018). Ecological correlates of Spanish preschoolers’ physical activity during school recess published in European Physical Education Review
Response 2: the indicated references were added, and additionally we have added 10 references at an international level and published in impact journals. Also, we reviewed some metanalysis and systematic review about the topic.
Point 3: Specific comments.
Abstract
Line 18. Please, state the sense of the differences so that the reader does not have to check the table to know which one of the groups presents the higher value.
Response 3: the sense of the differences were added in abstract
Line 39. Please, provide evidence support to this statement.
Response 4: statement supported (see comentary in the manuscript)
Line 52-65. The authors indicate different factors (i.e., SES, nutritional conditions and obesity) influencing children's psycho-motor development. The question is, why did the authors only focused on the obesity. Why did not the authors assessed the influence of SES and obesity together? Alternatively, why did not the authors adjust the analyses according to the SES?
Response 5: the project was planned from the beginning to analyze the effects of the BMI no information was collected on other variables such as nutritional status or socio-economic level, this is the reason why this type of variables were not analyzed
Line 66 (purpose). In relation to the main purpose of this manuscript, it is well addressed the importance of educational community in regards to binomioum sedentary behaviour-nutrition with consequences in children's weight status. However, it would be needed to enhance its rationale in consequences such as motor competence, cognition, perception and/or affective development.
In addition, why do the authors just analysed 5-years-old children? Why do not they analyse 3- 4- and 5-years-old?
In addition, in the current study, comparative analyses according to not only the BMI but also children's sex (5x2 model) are conducted. However, the rationale of the effect of sex in children's psycho-motor development is not clear enough. The question is, why do the authors assume there is differences between 5-years-old boys and girls.
Response 6: the three questions were adressed in different paragraphs of the introduction of the manuscript (please see comments)
Line 85. For those who are not familiar with the methods used, which anthropometric indicators are required and who do you implement it? Please, clarify.
Response 7: anthropometric indicators were added
Line 86. According to Table 1, this method seems to be based on descriptive values (SD). So, please, justify why having this method is more appropriate than BMI percentiles (adjusted for sex and age) with classification in normal, overweight and obese children (a method suggested by the CDC, 2000).
Response 8: tables used were based in BMI percentiles suggested by the CDC, this was cleared in the footnote of Table 1
There are acronyms without explanation in tables. For instance, in table 1, it seems the authors refer to IMC for BMI. If so, please, change "IMC" by "BMI". Check it throughout the manuscript.
Response 9. Acronyms were checked in all the tables.
Line 92-105. It can be accepted the CPA has been validated previously. However, it could be also argued why the authors consider a functional aspect such as respiratory control as perceptive. In addition, taking into account that the affective-relational dimension involves just emotional control and social relationships, why do not the authors separate these two aspects in two different dimensions: a) affective (by emotional control) and b) relationships (by social relationships)? These are examples of the necessity of describing this instrument thoroughly.
In addition, please, note that the explanation of the tasks assessed is rather ambiguous. It seems this is a proxy-report by the preschool teachers in relation to the motor, perceptive and affective children's development. As this is a new instrument, please, clarify. In the case, this is a proxy report, how do the authors assess children actual motor, perceptive and affective/relational development? The original reference neither explain it (Romero-Martínez et al. (2018). Perceptual and Motor Skills, 125(6) 1070–1092).
Response 10. Instrument and tasks that include was described thoroughly in the Table 2 (added, please view comments in the manuscript). Items of respiratory control was classified by the experts in the development of CPA as perceptive because involve an organ of the senses such as the nose.
Line 105. It seems the authors are using previous data published in the manuscript about the validity and reliability of the CPA scale with exact the same comparisons and method in Discriminant Validity (Romero-Martínez et al., 2018). If so, what is the novelty of the current study. Also, it should be stated. In the case, I am not right, please, clarify.
Response 11:the discriminant validity of the study by Romero et al. (2018) included two procedures, on the one hand the comparison between preterm and nonpreterm children and on the other the comparison of groups according to the BMI. The objective of this study was to present the evidences on the psychometric properties and the complete developmentof a new checklist-type instrument to evaluate the psychomotor development of children, which is a totally different objectivefrom the one presented here. In fact, the introduction of this article is based on the evaluation of psychomotricity and not on the study of the effects of overweight in the different variables that compose the psychomotor development. If you refer to table 7 of the article by Romero et al. (2018) please note that a separate analysis has not been done in boys in girls (in fact, the results are different). This table has been done in a global way and in order to prove the discriminant validity of the instrument (under a psychometric study)
Line 109. How were the classroom teachers instructed? A thorough description of this procedure is requested.
Response 12: a description of the instruction of teachers was added
Line 112. How did the researchers and teachers measure children's height and weight?
Response 13: added
Line 116. What do the authors mean when they say ex-post facto prospective simple design? As far as this reviewer recognise (according to the procedure explained) this is a cross-sectional study. Is there any further information that should be considered/included to define the study design?
Response 14:Changed inmanuscript
Line 123. In terms of what are the authors comparing the results of the youngest children? Please, clarify.
Response 15:Line 123 is the title of the section results, we don’t understand the question
Line 127. The explanation of the association of BMI and balance in young children (4-6 years-old) does not seem to be congruent enough. The authors state that in early childhood the BMI is not a factor that influence balance, however, they use data by Dinkel with babies of 5-6 months. I would suggest the authors delete the explanation in lines 128-132 by using lines 133-137 for explaining the absence of differences between normal and overweight early children.
Response 16:Line 127 is “The results of the Kruskal-Wallis H test for the motor skills according to the BMI in girls is presented in the table 2”, we don’t understand the question. In addition we don’t cite Dinkel in this manuscript.
Line 131. Why do the authors use LISREL and Prelis in the data analysis? Did the authors include any structural equation modelling for the analyses? Please, clarify.
Response 17:Changed in manuscript. We only used SPSS, in a draft version of the manuscript we included the analysis of a SEM model but it was removed in the final version.
Line 132. I mainly agree the authors justification of non-parametric analyses. However, taking into account the 5x2 comparison in terms of BMI and sex. Why did not the authors conduct a mix-model with these two factors included so that main and interaction effects could be test? This would enhance the potential of the analyses.
Response 18: our interest was to analyse girls and boys separately. ANOVA cant be used due to the breach of the assumptions.
Table 2. Please, remove the meaning from the text and clarify what does Me, G1 and G2 mean in the note below the table.
Response 19: both actions were made in manuscript
Moreover, why do the authors include an asterisk into brackets? What does this asterisk mean?
Response 20: It is an erratum
Line 135. I strongly recommend the authors to use a Figure to provide pairwise comparisons. The statistics of the post-hoc analysis could be reported in a supplemental file.
Response 21: pairwise comparisons were summarized in figures 1 to 4
Why do the authors provide the r-value? Is this for indicating the effect size? If so, how do the authors interpret this statistic? In addition, why do not the authors use another statistics such as V-value for non-parametric statistics?
Response 22: yes, it is a common measure of effect size in non parametric test, it is explained in the results section
Taking into account there are so many comparative output from data (at least values from 16 dependent variables were compared according to 5 levels of BMI for boys and girls) to be discussed, and the structure and content of the discussion, why do not authors just provide values and comparisons from the SUM of every aspect in the three dimension (motor, perceptive and affective) of the CPA scale?
Response 23: comparisons from the SUM of motor, perceptive and affective factors were included in discussion
Line 265. Please, add "for boys and girls" at the end of the sentence.
Response 24: yes, it is a common measure of effect size in non parametric test, it is explained in the results section
Lines 271-277. The authors are discussing the results for boys and girls, and for obese and non-obese interchangeable. As a result, sometimes it is not clear whether they refer to the factor sex or the BMI. It would be recommended that they clearly separate the discussion of both factors.
Response 25: this paragraph was centered only in gender comparison
Line 279. Lower than who? Please, clarify.
Response 26: Fixed in the manuscript please see comments
Line 282. Please, note that this is the first time the acronym (PhM) appears in the text.
Response 27: changed in the manuscript please see comments
Conclusions
Line 352. Please, rewrite the first conclusion. The sentence is rather large and requires improvement.
Response 28: re-elaborated in the manuscript (please see comments)
Line 359. This is not a conclusion derived from the results of the study directly. This is a secondary assumption by the authors that has been already stated in the discussion section. It is also repeated in the third conclusion.
Response 29: eliminated in the manuscript (please see comments)
Reviewer 3 Report
Dear Madrona et al
Thank you for your work on the submitted manuscript. It is in my opinion of good contemporary interest, and following revision will make a good addition to the literature .
Specific comments are to follow;
Line 14 - index is not plural .Change to indices
Abstract overall - good content, but English use and sentence structure is relatively poor and must be improved.
Introduction -
Again, as a whole this requires a strong edit in terms of English and sentence structure.
The use of "specially" is wholly incorrect, I believe "especially" is the word you require.
I feel th st the introduction, whilst in part was fine, requires greater expansion of motor control/competence concept, this in particular is underdeveloped.
Equally, given it's importance to the manuscript, on would expect the author's to better rationalise CPA and its constituents.
Table 1 - less than not less of, what is "muestra", define "IMC"
Instruments -
Greater expAnsion of the checklist is needed, including relevant validity and reliability metrics for the age group studied. I appreciate the link to relevant citation, but you must also provide a synopsis for three reader to make judgement on the suitability of your test choice.
Procedure
Greater specific detail of how height and weight were measured and using what.
Likewise, the pre-processing of data needs to be completely transparsnt, so that we know the steps you Have taken, and to facilitate reproduction as needed. So please report it all.
The results are comprehensive and clear
I do have a request however, given you discuss it partly, but you will likely need to run some statistical analysis on the male-female comparison. Without it, you don't have much rationale for it's inclusion inquire discussion.
Discussion
Throughout manuscript - change gender to sex - sex is the biological demarcation whilst gender is a sociological construct.
Line 271 - I take exception to your comparison with catenassi, you report that they reported no correlations, in fact, neither did you, so your comparison of correlation and difference testing is not sound. Please 're frame it for scientific accuracy.
Line305 - I do not believe your assertions are not well founded, given you did not conduct any sex difference testing.
Limitations -I disagree that non parametric stats equate to a weakness, I feel you have correctly treated your data, and the normality vs non-normality is negligible.
I feel you need to include a section or paragraph specifically highlighting the practical implications for teachers, practitioners and children. Also highlighting how this could be used to.inform interventions.
Author Response
Dear reviewer, thank you very much for the careful review of the article and the valuable suggestions made. Below you will find our answer to each point suggested.
Point 1. Line 14 - index is not plural .Change to indices
Response 1: Ok. Changed.
Point 2. Abstract overall - good content, but English use and sentence structure is relatively poor and must be improved.
Response2:English use and structure were improved.
Introduction -
Again, as a whole this requires a strong edit in terms of English and sentence structure.
Response3:Introduction was re-elaborated in order to improve English and sentence structure
The use of "specially" is wholly incorrect, I believe "especially" is the word you require.
Response4:Ok. Chaged. Please, see comments in manuscript.
I feel th st the introduction, whilst in part was fine, requires greater expansion of motor control/competence concept, this in particular is underdeveloped.
Response5: Paragraphs 2 to 7 of the introduction were re-elaboratedto include a conceptual discussion about definition, implementation, theories and components of psychomotricity and their relationships.
Equally, given it's importance to the manuscript, on would expect the author's to better rationalise CPA and its constituents.
Response6:sectioninstrumentwas re-elaborated in order to explain throughly the CPAand the actions/items that composed it (please see comments in manuscript).
Table 1 - less than not less of, what is "muestra", define "IMC"
Response7:Words in Spanish changed in this table(BMI and sample)
Instruments -
Greater expAnsion of the checklist is needed, including relevant validity and reliability metrics for the age group studied. I appreciate the link to relevant citation, but you must also provide a synopsis for three reader to make judgement on the suitability of your test choice.
Response8:sectioninstrumentwas re-elaborated in order to explain throughly the CPAand the actions/items that composed it (please see comments in manuscript).
Procedure
Greater specific detail of how height and weight were measured and using what.
Response9:explanation about how the weight and height were measured was included in the manuscript (please, see coments)
Likewise, the pre-processing of data needs to be completely transparsnt, so that we know the steps you Have taken, and to facilitate reproduction as needed. So please report it all.
Response 10:explanation about pre-processing of data was included in the finalparagraph of the procedure section
The results are comprehensive and clear. I do have a request however, given you discuss it partly, but you will likely need to run some statistical analysis on the male-female comparison. Without it, you don't have much rationale for it's inclusion inquire discussion.
Response 11: our interest was to analyse girls and boys separately. ANOVA cant be used due to the breach of the assumptions.
Discussion
Throughout manuscript - change gender to sex - sex is the biological demarcation whilst gender is a sociological construct.
Response 12: Ok, changed.
Line 271 - I take exception to your comparison with catenassi, you report that they reported no correlations, in fact, neither did you, so your comparison of correlation and difference testing is not sound. Please 're frame it for scientific accuracy.
Response 13:Comparison with Catenassi were eliminated and changed for other more accurate comparisons
Limitations -I disagree that non parametric stats equate to a weakness, I feel you have correctly treated your data, and the normality vs non-normality is negligible.
Response 14:we have qualified said limitation
I feel you need to include a section or paragraph specifically highlighting the practical implications for teachers, practitioners and children. Also highlighting how this could be used to.inform interventions.
Response 15:This paragraph was included in the manuscript (please, see comments)
Round 2
Reviewer 2 Report
It is expected that having the chance of enhancing the quality of a manuscript, authors are conducting a revision with this aim. In the introduction the rationale has been enhanced however, in sections such as methods and results this does not seem to be the case.
In spite of the lack of authors’ attention to some formal aspects, I am still considering this study as interesting. I would like to assume that, maybe due to the limited available time for conducting the revision, the authors did not address every formulated question appropriately:
1. It must be noted that there are many grammatical and typo errors. The English must be revised by a native speaker.
2. Citation style does not match with the number of references (see example below).
3. The method does not lead other researchers to replicate the study.
a. BMI. This reviewer feels it is very interesting how the authors compute and classify every child into the five levels of BMI, however, it is not explained in detail nor the citation corresponds with the number of reference.
b. CPA. A major issue is related to the test used for assessing psychomotricity. This is novel, however there is a lack of explanation in describing how the test is conducted thoroughly.
The protocol and how the instrument is applied in children are not well explained. Actually, the current version does not lead to replicate the study. For instance, meanwhile table 2 presents the tasks/actions every child has to do, it is not well explained how it is done.
Questions that exemplified this comment are as follows:
What are the instructions provided to the child?
How do you score every action? In other words, how do an assistant researcher score 1, 2, 3, 4 and/or 5 in every action?
What does every value mean?
What is the range of values for every variable and the CPA?
How long time did the procedure take to be administered?
These questions should be addressed in detail.
4. The results section does not help the readers to understand easily what this study found. The information is interesting enough, nonetheless the question is: is it needed to provide this huge amount of information for transmitting the idea of the study?
Due to the excessive length of the entire section the findings of this manuscript are not easy to understand. It would be strongly recommended authors just present the values for the factors and only the comparisons of variables in figures (e.g., interquartile range, bars, etc.).
5. New figures are not appropriate. Every figure and table must have sense by itself. It is required a thorough description of Figure 1. For instance, what does 1-5 mean? Pairwise comparisons in terms of what?
Moreover, the fact of including a figure is to summarize the information provided in the manuscript. It does not seem to have sense whether the post-hoc comparisons are still written in the main text. For instance, figure 2 style is awkward. Authors are encouraged to find a different style (e.g., interquartile range, bars, etc.). Please, take your time and try to find the best style to present your results.
6. It seems there is a gap between the way the results are presented and the discussion style.
7. Meanwhile the authors addressed some changes in the discussion, it is not homogenized through the text. Please, note that the sense of the differences is provided in some cases, but not in others. An example of this issue can be found in lines 215-246. Here it is still present the absence of the sense of the differences. The authors just say there are differences but it is not possible to know in this current form of explaining post-hoc comparison which group present higher or lower values.
8. There are issues in the references style (see for instance reference 24 with two studies cited).
9. In addition, please, note that for enhancing the authors/reviewers communication during the revision process, answers to the authors should be explained in more detail. It would be good to know exactly where every point has been changed in the manuscript [for further revision, please, state it (e.g., see lines XX-XX)]. For instance, the authors responded a model 5x2 to analyse the interaction between BMI and sex is not possible to be conducted however, they do not explain why it is not possible. There are possibilities for addressing this model however no justification is offered.
Author Response
It is expected that having the chance of enhancing the quality of a manuscript, authors are
conducting a revision with this aim. In the introduction the rationale has been enhanced however, in sections such as methods and results this does not seem to be the case.
In spite of the lack of authors’ attention to some formal aspects, I am still considering this study as interesting. I would like to assume that, maybe due to the limited available time for conducting the revision.
Response/ Yes, we had very little time considering that we were in Christmas holidays
The authors did not address every formulated question appropriately:
1. It must be noted that there are many grammatical and typo errors. The English must be revised by a native speaker.
2. Citation style does not match with the number of references (see example below).
3. The method does not lead other researchers to replicate the study.
a. BMI. This reviewer feels it is very interesting how the authors compute and classify every child into the five levels of BMI, however, it is not explained in detail nor the citation corresponds with the number of reference.
b. CPA. A major issue is related to the test used for assessing psychomotricity. This is novel, however there is a lack of explanation in describing how the test is conducted thoroughly. The protocol and how the instrument is applied in children are not well explained. Actually, the current version does not lead to replicate the study. For instance, meanwhile table 2 presents the tasks/actions every child has to do, it is not well explained how it is done.
Questions that exemplified this comment are as follows:
What are the instructions provided to the child?
How do you score every action? In other words, how do an assistant
researcher score 1, 2, 3, 4 and/or 5 in every action?
What does every value mean?
What is the range of values for every variable and the CPA?
How long time did the procedure take to be administered?
These questions should be addressed in detail.
Response/ Two paragraphs were added on page 4 that answer all these questions
The results section does not help the readers to understand easily what this study found. The information is interesting enough, nonetheless the question is: is it needed to provide this huge amount of information for transmitting the idea of the study? Due to the excessive length of the entire section the findings of this manuscript are not easy to understand. It would be strongly recommended authors just present the values for the factors and only the comparisons of variables in figures (e.g., interquartile range, bars, etc.). New figures are not appropriate. Every figure and table must have sense by itself. It is required a thorough description of Figure 1. For instance, what does 1-5 mean? Pairwise comparisons in terms of what?
Moreover, the fact of including a figure is to summarize the information provided in the manuscript. It does not seem to have sense whether the post-hoc comparisons are still written in the main text. For instance, figure 2 style is awkward. Authors are encouraged to find a different style (e.g., interquartile range, bars, etc.). Please, take your time and try to find the best style to present your results. It seems there is a gap between the way the results are presented and the discussion style.
Response/ The results section has been fully reworked, the previous figures have been eliminated and other new were included (in your first review we did not understand what kind of figures you wanted us to do) in the end we have chosen to make figures with the average ranges that are the statistics that are analyzed in the nonparametric test. The figures seem clear to us and are well explained in the text. The Mann-Whitney U test for post-hoc comparisons (pairwise) has been moved to an annex as you suggested in your first review.
Meanwhile the authors addressed some changes in the discussion, it is not homogenized through the text. Please, note that the sense of the differences is provided in some cases, but not in others. An example of this issue can be found in lines 215-246. Here it is still present the absence of the sense of the differences. The authors just say there are differences but it is not possible to know in this current form of explaining post-hoc comparison which group present higher or lower values.
Response/ Discussion were changed to improve homogeneity throughout the text.
There are issues in the references style (see for instance reference 24 with two studies cited).
Response/ All references were revised several times.
Reviewer 3 Report
Dear authors; thank you for your hard work in answering all of my queries. I feel that you have competently answers all of queries and concerns; I only have 2 comments. 1) I suggest removing the pairwise comparison figures, I do not believe they add anything, and look visually poor (because they arent needed). Nevertheless, I leave this up
To the editor. 2) a very tight grammar and sentence structure edit is needed prior to acceptance in my view.
Overall, 2 very easy to remedy points, and I would then gladly suggest publication. Great work.
Author Response
Dear authors; thank you for your hard work in answering all of my queries. I feel that you have competently answers all of queries and concerns; I only have 2 comments. 1) I suggest removing the pairwise comparison figures, I do not believe they add anything, and look visually poor (because they arent needed). Nevertheless, I leave this up to the editor.
R/ Ok. the previous figures have been eliminated and other new were included. The figures seem clear to us and are well explained in the text.
2) a very tight grammar and sentence structure edit is needed prior to acceptance in my view.
R/ Ok, the manuscript was reviewed by a native translator in English